# Experimental study of precursory features of $CO_2$ blasting-induced coal rock fracture based on grayscale and texture analysis

**Hongyu Pan, Haotian Wang[ID]\*, Kang Wang, Tianjun Zhang, Bing Ji**

College of Safety Science and Engineering, Xi'an University of Science and Technology, Xi'an, China

\* wht@stu.xust.edu.cn

**Data Availability Statement:** All relevant data are within the manuscript and its Supporting Information files

## Abstract

$CO_2$ blasting has been identified as a potent method for enhancing the permeability of coal seams and improving gas drainage efficiency. This study is focused on elucidating the deformation and fracture mechanisms of coal and rock during $CO_2$ blasting and on identifying the precursor characteristics of these processes. To this end, a $CO_2$ blasting-induced coal rock fracture pressure model and a gas pressure distribution model were developed. The research utilized a self-developed $CO_2$ blasting test platform along with a non-contact full-strain field measurement analysis system. Briquette samples were subjected to $CO_2$ blasting tests under controlled experimental conditions, which included an axial pressure of 1.0 MPa and variable gas pressures of 0.5, 1.0, and 1.5 MPa. This methodology enabled the capture of the principal strain field on the surface of the samples. The Gray Level Co-occurrence Matrix (GLCM) was employed to extract and analyze the grayscale and texture features of the strain cloud maps, facilitating a quantitative assessment of their evolution. The aim was to pinpoint the precursor characteristics associated with coal rock cracking and crack propagation. The results revealed that: (1) During the cracking and subsequent propagation of samples, the strain field's grayscale histogram underwent a transformation from a "broad and low" to a "narrow and high" configuration, with a consistent increase in peak frequency. Specifically, at 3 ms, a primary crack was observed in the sample, evidenced by a grayscale peak frequency of 0.0846. By 9 ms, as the crack propagated, the grayscale peak frequency escalated to 0.1626. (2) The texture feature parameters experienced their initial abrupt change at 3ms. Correlation with the gas pressure distribution model indicated that this was the crack initiation moment in the sample. (3) A secondary abrupt shift in the texture feature parameters occurred at 9ms, in conjunction with experimental phenomena, was identified as the crack propagation phase. Monitoring the grayscale and texture features of the principal strain field on the coal rock surface proved effective in recognizing the precursor characteristics of crack initiation and propagation. This research has the potential to reduce blasting costs in coal mines, optimize blasting effects, and provided theoretical guidance for enhancing gas extraction efficiency from deep and low permeability coal seams.

**Funding:** This research was financially supported by The National Natural Science Foundation of China (52274226). The funder had important role in writing – review & editing of the manuscript.

**Competing interests:** The authors have declared that no ompeting interests exist.

## Introduction

The high gas content in China's coal seams significantly hinders the safe and efficient production of mines [1]. Gas drainage is the most economically effective technical method to reduce coal seam gas content and achieve the standard of gas drainage in mines [2]. Scholars have extensively researched methods to enhance coal seam permeability, increase gas extraction efficiency, and reduce pre-drainage time. Techniques like hydraulic fracturing [3], hydraulic slotting [4], and deep hole pre-splitting blasting [5] have been explored. Yet, these methods have inherent limitations due to their technical mechanisms and construction processes. For instance, hydraulic methods may lead to water wastage and adversely impact the working environment [6], while deep hole pre-splitting blasting risks generating heat, potentially causing secondary explosions. $CO_2$ blasting, on the other hand, shows superior qualities with its stable physical and chemical properties, ease of recyclability, and strong penetration capabilities [7]. It notably outperforms hydraulic fracturing in enhancing fracture flow capacity [8]. Lu et al. [9] found that the fracture initiation pressure in shale fracturing with $SC\text{-}CO_2$ was about 50.9% lower than that in hydraulic fracturing, and around 57.1% lower for sandstone. $CO_2$ blasting can create a complex fracture network and avoid the water blocking effects, with fracture apertures observed in $CO_2$ blasting being significantly larger than those in hydraulic fracturing, reaching up to 2–5 times larger under the same test conditions. Permeability tests indicated that the permeability from $CO_2$ blasting is 190.43 times that of hydraulic fracturing [10]. Additionally, $CO_2$ can displace $CH_4$ adsorbed in coal seams [11], enhancing gas drainage efficiency while enabling permanent $CO_2$ sequestration [12], thus holding significant potential for improving gas extraction efficiency from deep and low permeability coal seams. [13, 14].

In determining the initiation pressure of $CO_2$ blasting to enhance blasting and gas extraction effects, extensive research has been conducted. Wang et al. [15] constructed a mathematical model to determine the fracturing pressure and range of influence of $CO_2$ fractured coal seams. Guo et al. [16] established a flow-stress-damage coupled crack initiation and propagation model, and studied the factors affecting the efficiency of $CO_2$ fracturing. Yang et al. [17] compared the effects of hydraulic fracturing and $CO_2$ fracturing, and the experimental results showed that $CO_2$ fracturing had lower initiation pressure and produced a more complex network of fractures. Fan et al. [18] proposed a new type of liquid $CO_2$ fracturing and permeability enhancement technology, and established the calculation models of the initiation pressure and flow. Zhang et al. [19] proposed that monitoring parameters such as the initial gas blasting pressure and surface strain of the coal-rock mass must be obtained throughout the entire $CO_2$ fracturing process. Wang et al. [20] established a radial vibration mechanics model of $CO_2$ blasting, and used radial vibration characteristic parameters to divide $CO_2$ blasting into three stages. Cai et al. [21] studied the influence of coal seam joints on crack morphology and perforation damage under $SC\text{-}CO_2$ fracturing, and analyzed the stress-strain characteristics, internal crack propagation, and macroscopic crack failure modes of sandstone through experimental research on the relationship between flow field and strain change [22]. Gao et al. [23] studied the influence of $CO_2$ initiation pressure and shear plate thickness on $CO_2$ phase change fracturing and rock breaking process based on strain monitoring and fracturing pit volume measurement. Shang et al. [24] conducted $CO_2$ blasting experiments under true triaxial stress, and obtained the crack propagation law through monitoring the surface strain of the specimen. Huang et al. [25] conducted uniaxial compression tests to study the evolution of strain field, acoustic emission count, and local strain during the rock failure process. Wang et al. used fluid motion equations to construct analytical solutions for gas pressure attenuation at different positions [26], and inverted the energy field of the entire $CO_2$ blasting process through strain field inversion [27]. However, current characterizations of coal rock fracture

states mainly rely on experimental characterizations, such as CT scanning [28], nuclear magnetic resonance [29], 3D morphological scanning [17], and acoustic emission [30], with a lack of quantitative analysis methods. In subsequent analysis, it is necessary to quantitatively study the evolution characteristics of strain field in coal rock fracture states. To investigate the deformation and fracturing mechanism of fractured rock masses, Zhang et al. [31] conducted uniaxial compression experiments and studied the evolution characteristics of strain field grayscale and texture during the deformation and fracturing process of fractured sandstone specimens. Ji et al. [32] studied the gray images and gray features parameters of the coal and rock bodies around boreholes before and after grouting sealing during progressive damage, the crack propagation law was analyzed from the perspective of digital images. The research found that image features such as grayscale and texture can quantitatively characterize the evolution characteristics of strain field during the cracking and propagation process of rocks. The GLCM is a commonly used method to analyze image texture features. This paper used the GLCM to process the strain field cloud maps during the blasting process. It describes the texture features of the image by counting the spatial relationships between different gray values in the image. The GLCM can effectively describe the texture features of images, and has a good effect on some image processing tasks that need to consider the texture features. Currently, there is little research on the grayscale and texture features of strain fields in $CO_2$ blasting, and the research on identifying visible precursory features during coal rock cracking and propagation moments is insufficient.

In this context, we propose a new method based on image feature analysis to quantify the characteristics of the $CO_2$ blasting-induced strain field in coal rock. Therefore, a model for the initiation pressure of $CO_2$-induced fracturing in coal rock was constructed, and $CO_2$ blasting experiments were conducted on a self-developed $CO_2$ blasting test platform. The full-field measurement and analysis system of non-contact strain field was used to obtain strain cloud maps. The grayscale and texture feature parameters of the strain field were quantitatively calculated, and the evolution laws of the grayscale and texture features during the fracturing process were studied to identify the precursory features of coal rock cracking and propagation. The objective of this study is not only to improve coal seam permeability and gas extraction efficiency but also to foster the practice of sustainable development and the circular economy. $CO_2$ blasting, as an eco-friendly technique, effectively utilizes resources while aiding in the reduction of greenhouse gas emissions. In line with the United Nations Sustainable Development Goals, our research highlights the potential of innovative technologies in the energy sector for environmental protection and efficient resource use. Moreover, drawing from research on urban mining [33] and waste resource utilization [34–36], our methodology aligns with circular economy principles, aimed at reducing environmental pollution and resource waste through efficient resource recovery and utilization.

## Theory

### $CO_2$ blasting-induced coal rock fracture pressure model

According to the criterion of rock tensile fracture, when the circumferential stress of the borehole reaches the tensile strength of the rock, the rock will fracture, that is, the fracture condition of the rock is $\sigma_\theta = \sigma_t$, and the pressure at this time is the rock's fracture pressure [37]. The mechanical model is shown in Fig 1.

In the $(x,\theta)$ polar coordinates, the calculation formula for the borehole hoop stress is:

$$\sigma_\theta^1 + \sigma_\theta^2 = \sigma_t \tag{1}$$

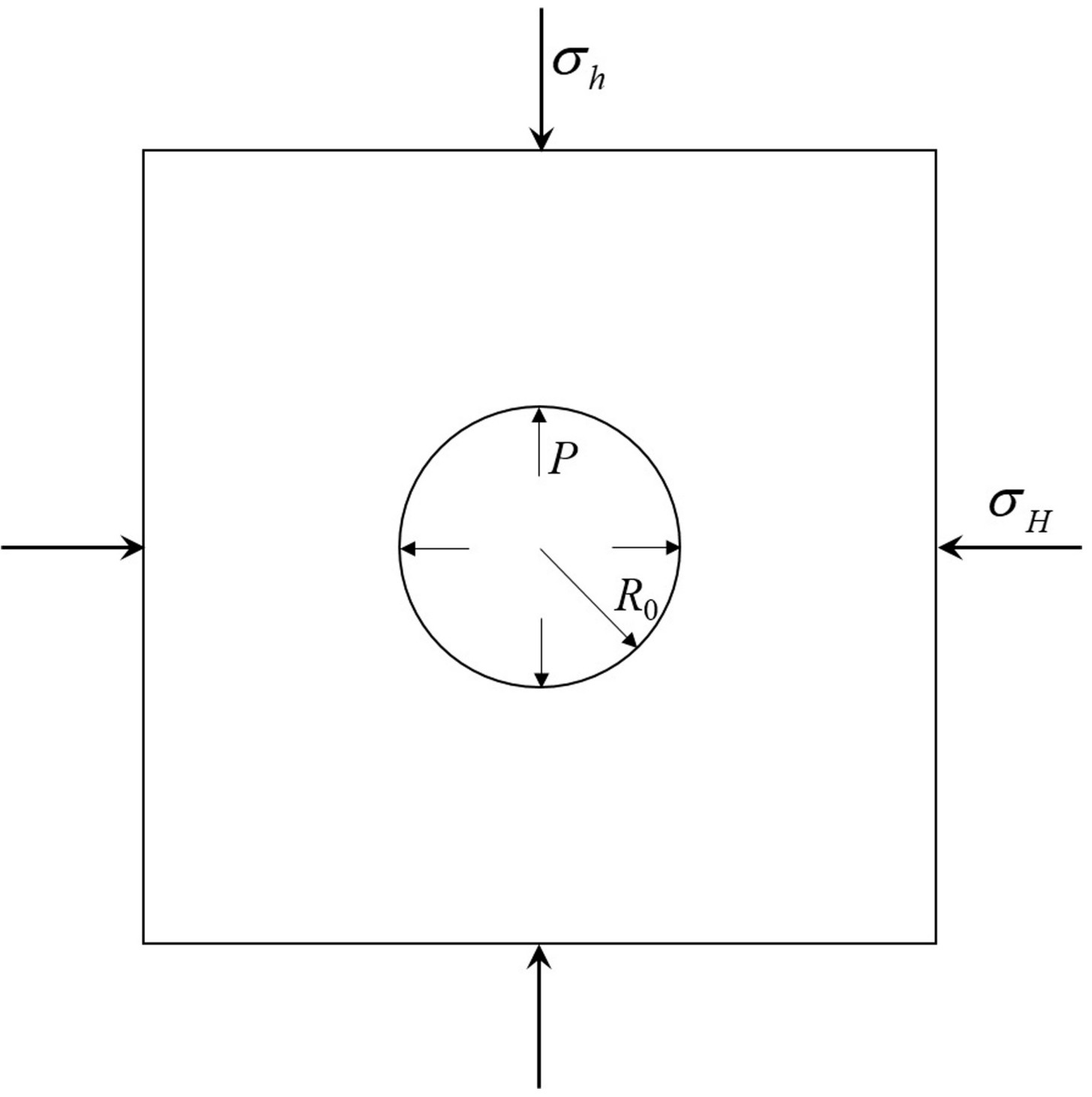

**Fig 1. The mechanical model.**

where $\sigma_\theta^1$ and $\sigma_\theta^2$ are as follows:

$$\sigma_\theta^1 = \frac{\sigma_H + \sigma_h}{2}\left(1 + \frac{R_0^2}{x^2}\right) - \frac{\sigma_H - \sigma_h}{2}\left(1 + 3\frac{R_0^2}{x^2}\right)\cos(2\theta) \tag{2}$$

$$\sigma_\theta^2 = \frac{R_0^2}{x^2}P \tag{3}$$

where $\sigma_\theta^1$ is the circumferential stress caused by the maximum horizontal principal stress $\sigma_H$ and the minimum horizontal principal stress $\sigma_h$, MPa; $\sigma_\theta^2$ is the circumferential stress caused by fluid pressure in the hole, MPa; $\theta$ is the circumferential Angle,°;$\sigma_t$ is the tensile strength of rock, MPa; $x$ is the radial distance to the stomatal center, m; $R_0$ is the radius of the borehole, m; $P$ is the pressure caused by the expansion and extrusion of the gas inside the borehole, MPa.

The above formula does not consider filtration, $x = R_0$, $\sigma_\theta^1$ value is related to θ, and θ = 0 or π is maximum. Substitute to obtain the crack initiation pressure $P_f'$, MPa:

$$P_f' = \sigma_t + \sigma_H - 3\sigma_h \tag{4}$$

Coal is a porous medium and $CO_2$ has strong permeability, so the starting pressure is affected by pore pressure. The additional circumferential stress caused by permeation is expressed as:

$$\sigma_\theta^3 = \alpha\left(\frac{1 - 2\mu}{1 - \mu}\right)\left[\frac{\int_0^r pxdx}{r^2} - p\right] \tag{5}$$

where $\sigma_\theta^3$ is the circumferential stress caused by $CO_2$ infiltration, MPa; $\mu$ is Poisson's ratio; $p$ is the pore pressure generated by $CO_2$ infiltration, MPa; $\alpha$ is the effective stress coefficient, and its value is between $0 \sim 1$.

According to Terzaghi effective stress theory [38], Eq (1) is transformed into:

$$\sigma_\theta^1 + \sigma_\theta^2 + \sigma_\theta^3 = \sigma_t - P \tag{6}$$

At the hole wall ($x = R_0$), Eqs (2), (3) and (5) are substituted into Eq (6) to obtain the crack initiation pressure $P_f$, MPa:

$$P_f = \frac{\sigma_t + \sigma_H - 3\sigma_h}{2 - \alpha\frac{1-2\mu}{1-\mu}} \tag{7}$$

## Distribution of gas pressure during $CO_2$ blasting

To investigate the relationship between the $CO_2$ gas pressure and time inside the borehole during $CO_2$ fracturing, the following assumptions were made: (1) The environmental temperature remains constant at the moment when the high-pressure gas from the liquid $CO_2$ phase change is ejected through the gas blasting tube. (2) The gas inside the borehole satisfies the ideal gas state Eq (3) The volume of gas inside the borehole remains constant before coal and rock mass failure, and at the beginning of the blasting process, the liquid $CO_2$ undergoes a phase change to a supercritical state, where its intermolecular forces are similar to those of a gas due to its nonpolar molecular structure. Therefore, the *R-K-S* equation is adopted for

calculation [39]. The expression of the *R-K-S* equation is as follows:

$$P = \frac{RT}{V_m - b} - \frac{a(T)}{V_m(V_m + b)} \tag{8}$$

$$b = 0.08664 \frac{RT_c}{P_c} \tag{9}$$

$$a(T) = 0.42748 \frac{R^2 T_c^2}{P_c} \left[1 + m'\left(1 - T_r^{0.5}\right)\right]^2 \tag{10}$$

$$m' = 0.48 + 1.574\omega - 0.176\omega^2 \tag{11}$$

Where $P$ is the gas pressure, Pa; $P_c$ the critical pressure, Pa; $V_m$ is the molar volume of the gas, L/mol; $T$ is the gas temperature, K; $T_c$ is the critical temperature, K; $T_r$ is the reduced temperature, it is the ratio of the current thermodynamic temperature $T$ to the critical temperature $T_c$ of the gas; $R$ is the gas constant, 8.314J/(mol·K); $\omega$ is the acentric factor.

According to $V_m = \frac{M}{\rho}$,, where $M$ is the molecular weight of $CO_2$ gas, $\rho$ is the density of $CO_2$, kg/m$^3$, and the *P-T-$\rho$* state equation of $CO_2$ is:

$$P = \frac{\rho RT}{M - b\rho} - \frac{\rho^2 a(T)}{M^2 + b\rho M} \tag{12}$$

The flow rate of $CO_2$ at the pore mouth is given by:

$$q_0 = YSP_0 \sqrt{\frac{M\gamma}{RT}\left(\frac{2}{\gamma + 1}\right)^{\frac{\gamma+1}{\gamma-1}}} \tag{13}$$

The value of $Y$ is determined by the ratio of the initial pressure $P_0$ to the pressure inside the borehole $P_1$.

$$Y = \begin{cases} 1, \frac{P_1}{P_0} \leq \left(\frac{2}{\gamma+1}\right)^{\frac{\gamma}{\gamma-1}} \\ \left(\frac{P_1}{P_0}\right)^{\frac{1}{\gamma}}\left[1 - \left(\frac{P_1}{P_0}\right)^{\frac{\gamma-1}{\gamma}}\right]^{\frac{1}{2}}\left[\left(\frac{2}{\gamma-1}\right)\left(\frac{\gamma+1}{2}\right)^{\frac{\gamma+1}{\gamma-1}}\right]^{\frac{1}{2}}, \frac{P_1}{P_0} \geq \left(\frac{2}{\gamma+1}\right)^{\frac{\gamma}{\gamma-1}} \end{cases} \tag{14}$$

In the equation, $q_0$ represents the flow rate of $CO_2$, kg/s; $S$ is the area of the gas pore, m$^2$; $P_0$ is the initial pressure, Pa; $\gamma$ is the adiabatic exponent of the gas, which for a triatomic molecule has a value of 1.32.

The density of $CO_2$ inside the pore can be expressed as:

$$\rho = \frac{3\int_0^t q_0 dt}{4\pi R_0^3} \tag{15}$$

By combining Eqs (12)—(15), the relationship between internal gas pressure and time during the $CO_2$ blasting process can be obtained.

When the initial gas pressure is 0.5, 1.0, and 1.5 MPa, the relationship between the internal pressure $P_1$ and time is shown in Fig 2. $P_1$ increases to around 5 MPa and then tends to stabilize. This is because the gas density in the borehole reaches its peak, and the pressurization effect is not significant, indicating that the rock tends to fracture. When $P_0 = 1.5$ MPa, the pressure tends to stabilize after 3 ms, indicating the time for rock fracturing.

## Materials and methods

### Experiment system

To conduct $CO_2$ blasting experiments under different initial gas pressure conditions, a $CO_2$ blasting test platform was devised and constructed. This platform encompasses three integral components: a loading system, a blasting system, and an observation system. The loading system employs a DNS-200 electronic universal testing machine, which is tasked with determining the basic physical and mechanical properties of the specimen prior to experimentation. Additionally, it applies a consistent axial compression to the specimen at the outset of the experiment. The $CO_2$ blasting system is composed of an assembly including a $CO_2$ cylinder, a pressure reducing valve, a high-pressure solenoid valve, a pressure gauge, and an explosion-proof inflation device. This latter device is specifically designed for the controlled release of a constant pressure to induce fracturing of the specimen at the moment of blasting. For observation, the setup incorporates a non-contact full-field strain measurement and analysis system, which allows for the monitoring of the specimen's strain field. The data pertaining to the strain field is subsequently acquired through comprehensive post-processing. The configuration and components of the experimental system are illustrated in Fig 3.

### Sample preparation

Due to the large number of microcracks in the notched specimen, it is difficult to conduct single-factor experiments. Core drilling is difficult and cannot guarantee the homogeneity of the specimens. However, the mechanical properties of coal powder, cement, and gypsum composite materials are similar to those of coal and the cost is low. Pre-experiments showed that coal powder, cement, gypsum powder, and water in a mass ratio of 1:2:1:0.8 can fit well with the gas blast tube and is not prone to gas leakage or other issues during the blasting process. Therefore, this ratio was used to prepare the specimens. Firstly, the materials were mixed uniformly according to the ratio and poured into a 70×70×140 mm rectangular mold. After oscillation, a 10 mm$\phi$ blasting tube was placed at the center of the mold. There were two 2 mm$\phi$ vent holes at the end of the tube. The specimen was demolded after 30 min, and then cured for 28 d in a constant temperature and humidity box at a temperature of 19 ∼ 24°C and relative humidity of 20% ∼ 30%. Finally, the surface of the specimen was polished and trimmed, and the position of the specimen and the gas blast tube is shown in Fig 4.

Before the experiment, uniaxial compressive strength test was conducted on the samples to obtain their basic physical and mechanical properties. The physical properties are listed in Table 1.

### Experiment design

Previous research [40, 41] has established that an increase in fluid pressure can induce turbulence and other phenomena in the pore structure of coal and rock masses, significantly impacting the fracturing effect. To determine the fracture timing in coal and to analyze the strain field characteristics under different initial gas pressure conditions during blasting, three sets of samples were prepared. These were subjected to $CO_2$ blasting experiments at initial pressures

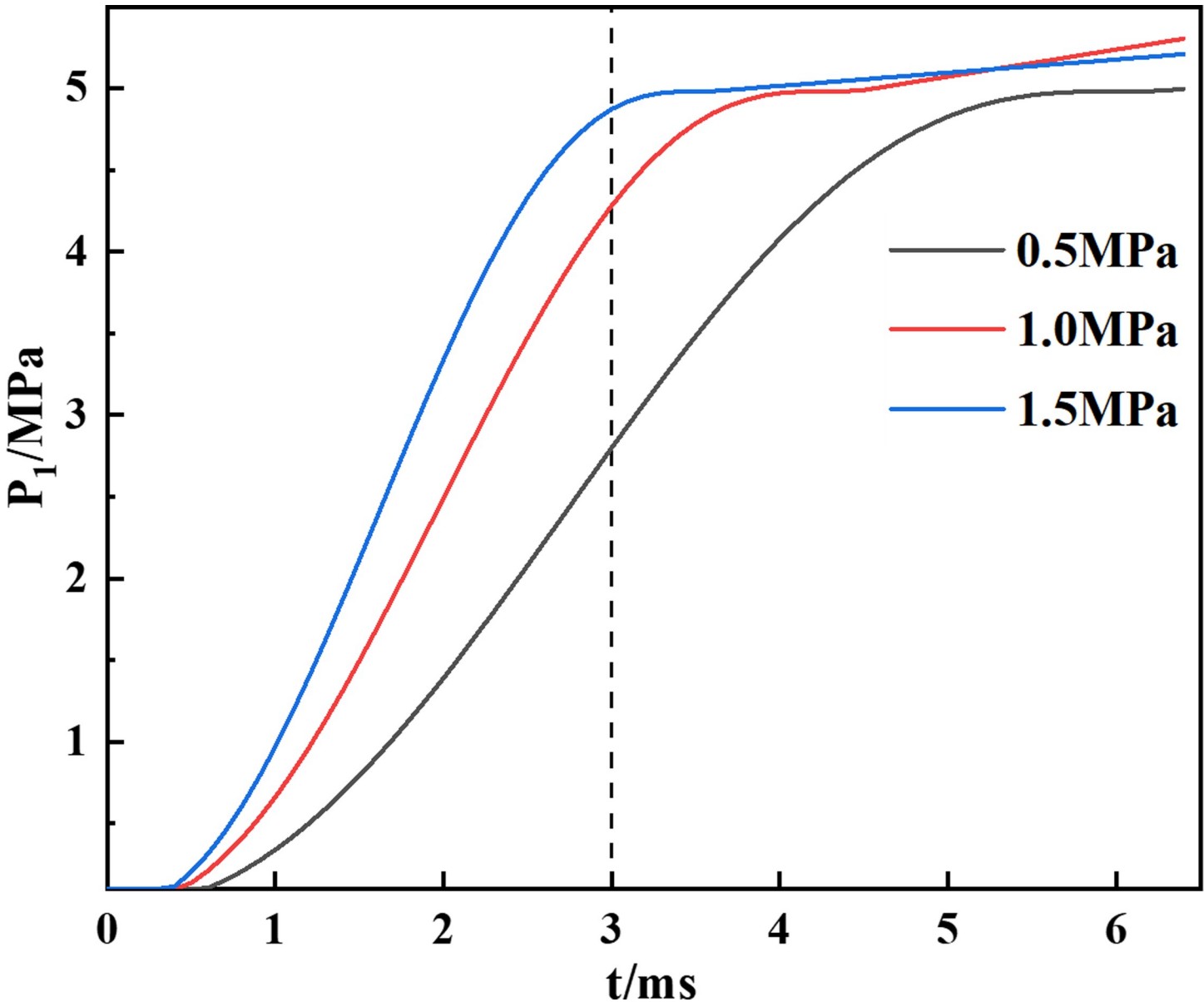

**Fig 2. Relationship between pressure and time under different initial gas pressures.**

of 0.5, 1.0 and 1.5 MPa, respectively. A non-contact full-field strain measurement and analysis system was employed to capture images of the region depicted in Fig 4 during the blasting process. The sampling interval was set at 1 ms to monitor the dynamic behavior of the strain field. Each set of experiments was replicated at least five times, with typical results being selected for in-depth analysis.

Prior to experimentation, the sample surfaces were manually prepared by applying white background paint and random black speckles to enhance image texture. Vaseline was used on the top and bottom surfaces to mitigate the end effect during testing. The loading process was force control, with a loading speed of 0.05 kN/s, appropriate for the low strength of the samples. The target load was set at 2.0 kN, achieving a constant pressure of 1.0 MPa during the

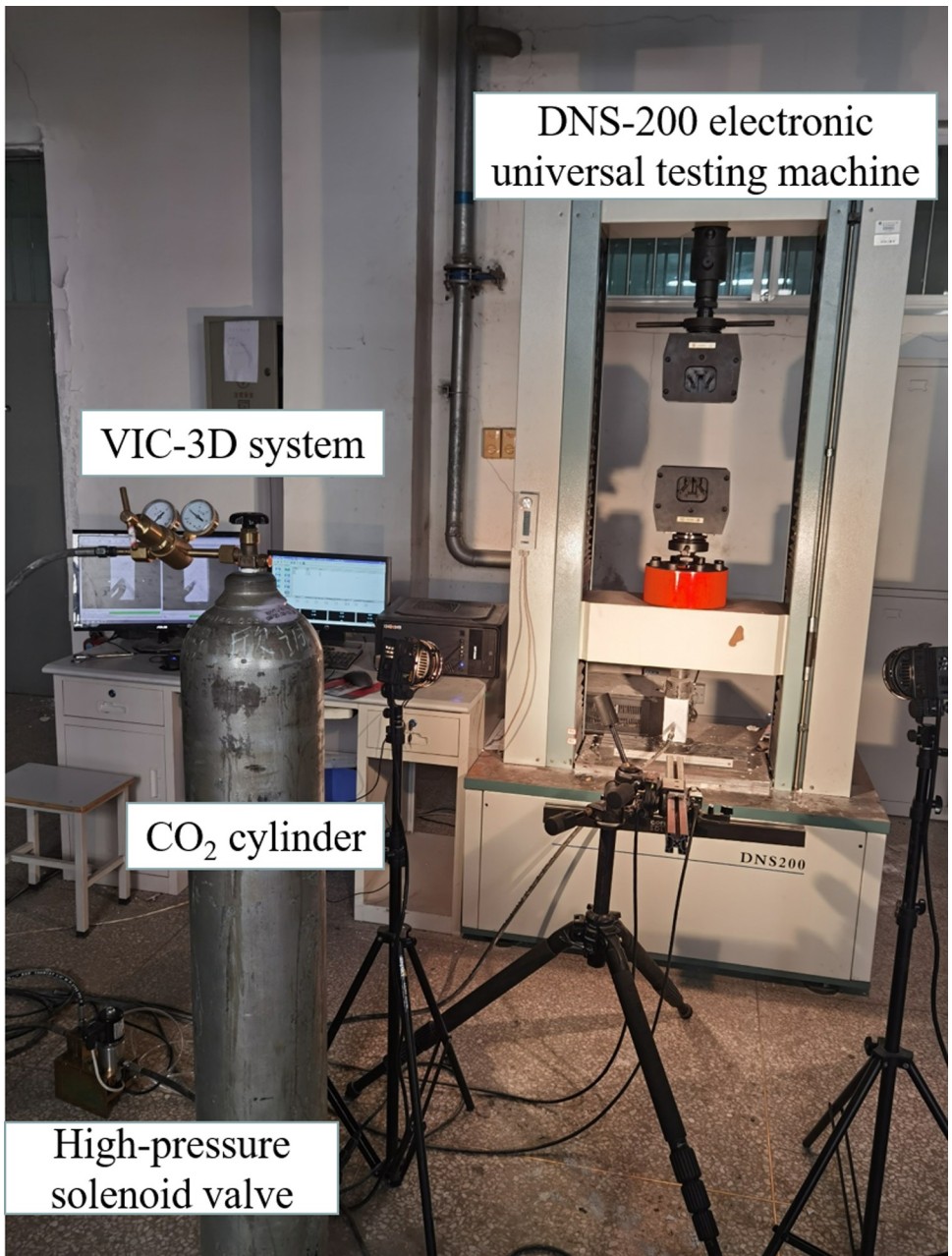

**Fig 3. $CO_2$ blasting test system.**

$CO_2$ blasting experiments under different initial gas pressure conditions. The experimental procedure was as follows:

1. A white light source illuminated the sample surface, and a camera captured the surface image. Focus, brightness, and other parameters were manually adjusted to ensure image clarity. The loading system and image acquisition system were synchronized to guarantee temporal consistency of data collection during the experiment.

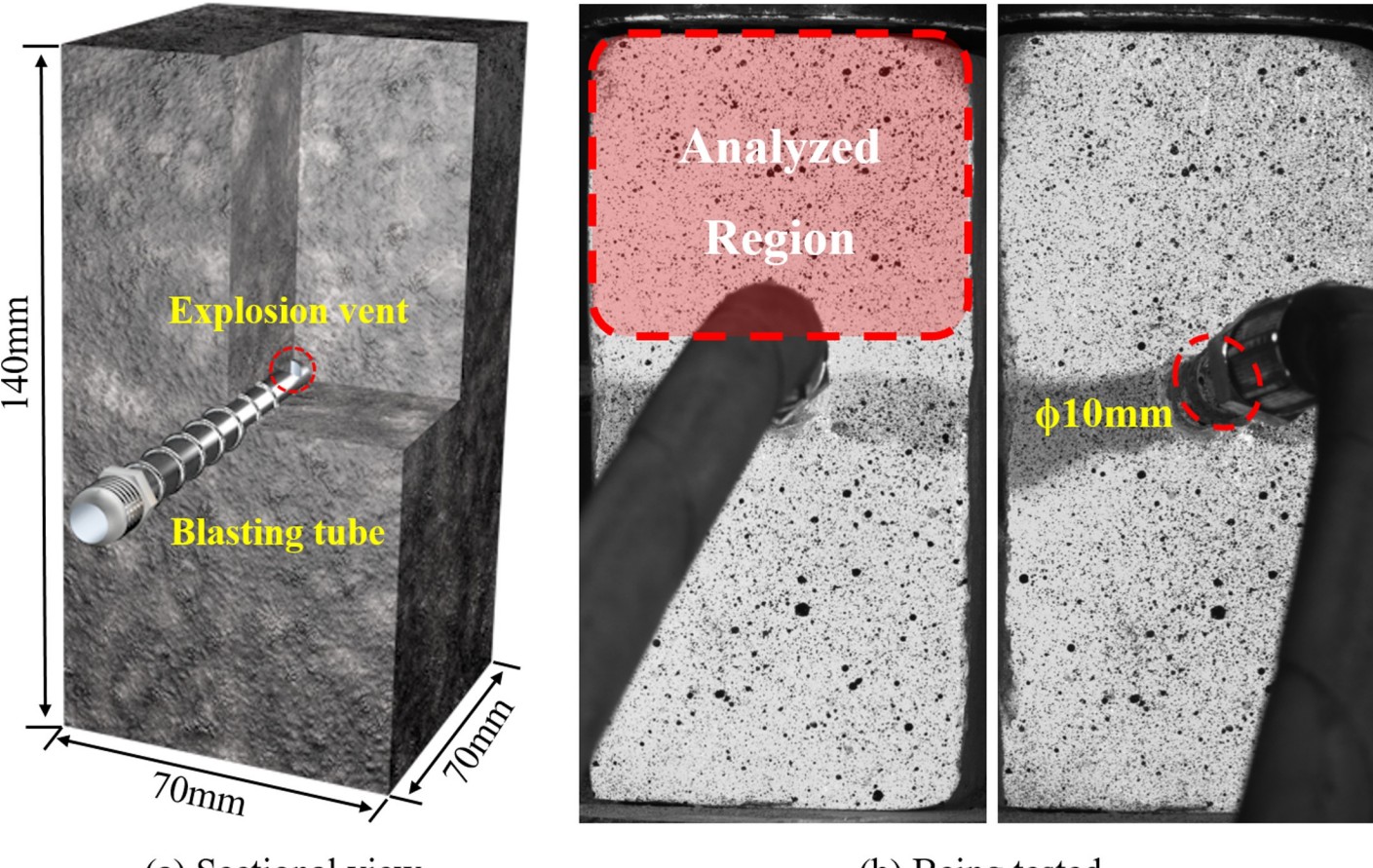

(a) Sectional view    (b) Being tested

**Fig 4. Schematic diagram of sample and position of gas blast tube.**

2. The pressure regulator was adjusted to maintain a constant pressure, monitored via the pressure display.

3. The high-pressure solenoid valve was activated to rapidly introduce $CO_2$ gas into the sample.

4. At the moment of the blasting, the non-contact full-field strain measurement and analysis system captured speckle images from the sample surface. The acquisition rate was 1 image/ ms, allowing for real-time monitoring of the strain field information until the sample incurred damage.

5. MATLAB-based custom were used for data analysis and processing, including:

   a. Generating principal strain cloud images of the specimen surface during the blasting process.

**Table 1. Physical and mechanical parameters.**

| Elasticity modulus/GPa | Shear modulus/GPa | Uniaxial compressive strength/MPa | Density /kg·m⁻³ | Poisson's ratio | Cohesion /MPa | Internal friction angle /˚ |
|---|---|---|---|---|---|---|
| 2.50 | 1.008 | 4.59 | $1.25 \times 10^3$ | 0.24 | 1.21 | 24.2 |

b. Converting these images into grayscale images, compressing the grayscale from 256 to 64 levels to reduce computational load.

c. Computing the grayscale histogram and texture feature parameters.

### Image analysis methodology

The workflow diagram of the analysis methodology is shown in Fig 5.

**(1) Gray histogram.** The VIC-3D software calculates the maximum principal strain field from the specimen surface image taken by the VIC-Snap system. The color scale was converted to 0–255 grayscale $k$ by Eq (16) [42].

$$k = 0.299R + 0.587G + 0.114B \tag{16}$$

In digital image processing, grayscale is a commonly used image feature, which can be used to describe the brightness, contrast and texture of the image. The grayscale histogram reflects the relationship between a certain grayscale level and its frequency of occurrence in the image. If the size of the image neighborhood window is $a \times b$, the mathematical expression is as follows [43]:

$$P(k) = \frac{n_k}{ab} \tag{17}$$

$P(k)$ represents the probability of $k$ grayscale level occurrence, $k$ is the grayscale level ranging from 0 to 255, and $n_k$ is the number of pixels with the grayscale level $k$.

**(2) Texture feature parameters.** There are many methods for extracting texture features [44], and currently, GLCM is recognized by the academic community as a robust and adaptive

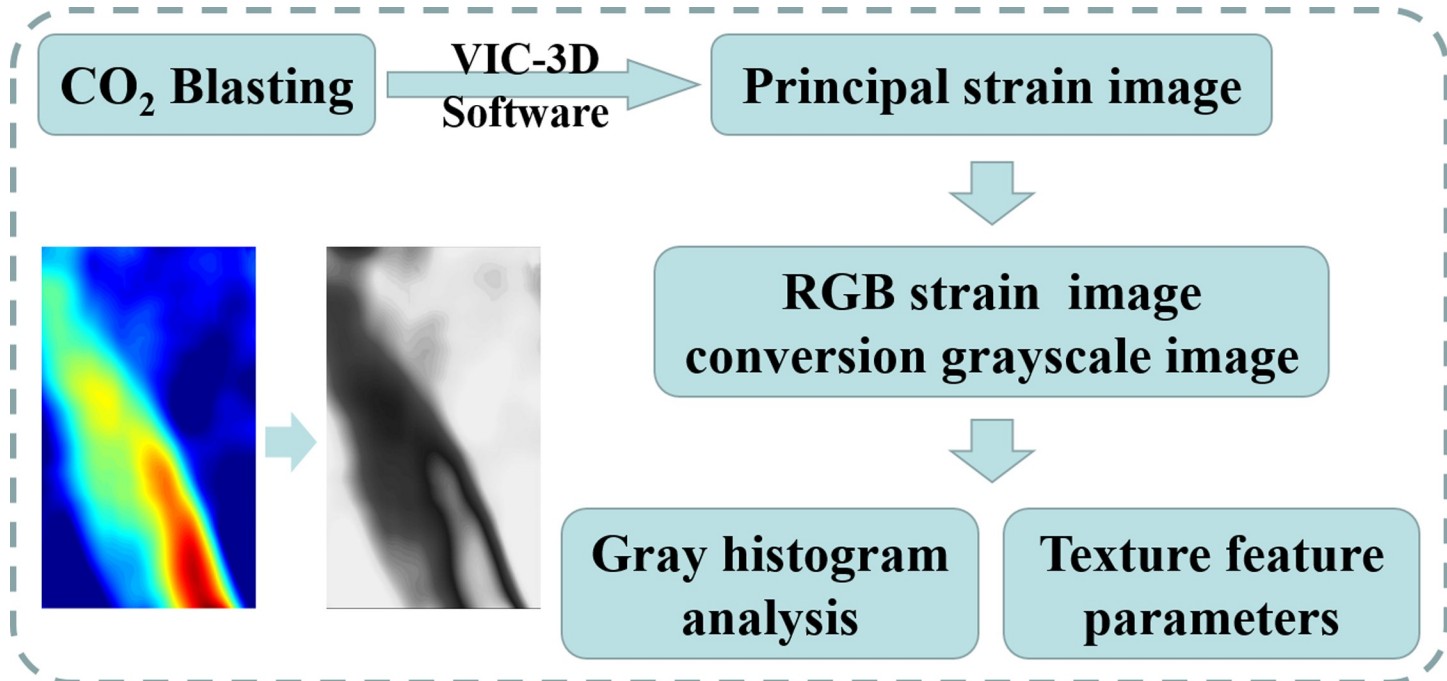

**Fig 5. Workflow diagram.**

theory. Moreover, it is a commonly used method for studying the spatial relationship of image pixel values. Therefore, this paper employs the GLCM to extract texture features.

The GLCM represents the probability $P(i,j,h,\theta)$ of having a pixel with gray level $i$ and another pixel with gray level $j$ at a distance of $(dx,dy)$. As shown in Fig 6. The mathematical expression is as follows:

$$P(i,j,h,\theta) = \{[(x,y),(x+\Delta x, y+\Delta y)]f(x,y) = i, f(x+\Delta x, y+\Delta y) = j\}$$
$$x = 0,1,2,\ldots,N_x - 1; y = 0,1,2,\ldots,N_y - 1 \tag{18}$$

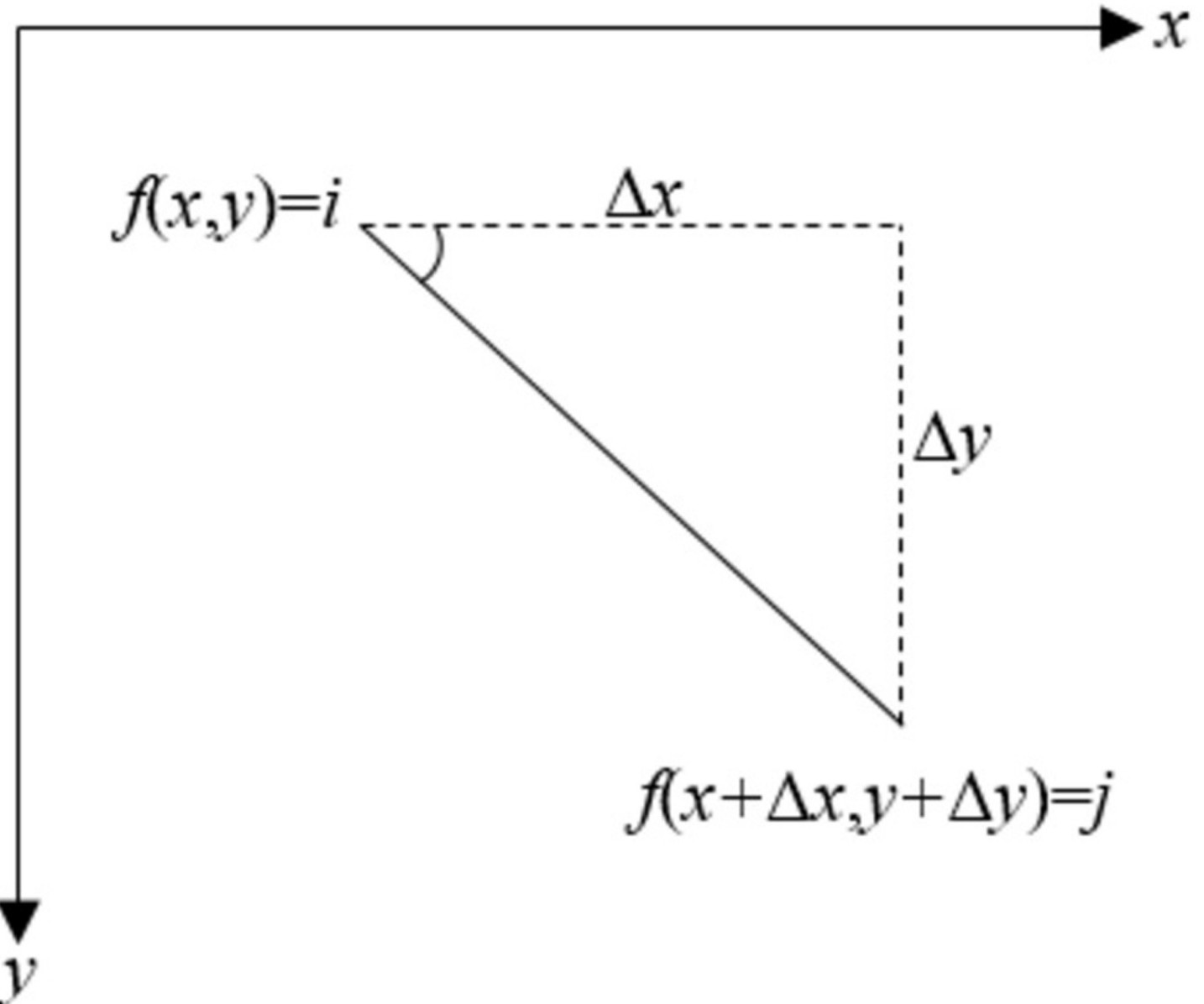

**Fig 6. The basic principle of GLCM.**

In the formula, $i$ and $j$ represent the gray levels ranging from 0, 1, 2, . . ., to 255; $x$ and $y$ are the coordinates of the pixels in the image; $N_x$ and $N_y$ denote the number of rows and columns in the image, respectively.

Based on the GLCM, the texture features information of the image can be accurately quantified [45]. Haralick et al. [46] extracted 14 feature values using the GLCM, among which, four feature values–angular second moment (ASM), contrast (CON), correlation (COR), and entropy (ENT) are easy to extract and are unrelated to each other. In this paper, the texture feature parameters of the strain field cloud map are extracted using the formulas shown in Table 2:

## Results and discussion

### Evolution law of strain field and gray histogram

The blasting tube is located at the centroid of the specimen, and the analysis of the $CO_2$ blast strain field characteristics is performed on the upper half of the specimen. A bespoke MATLAB program is utilized for transforming the principal strain field cloud map into a grayscale image. This program also plots the corresponding grayscale histogram and normalizes the data. During $CO_2$ blasting at an initial gas pressure of 1.5 MPa, the principal strain field cloud maps and grayscale histogram of the specimen are illustrated in Fig 7.

As depicted in Fig 7(A), at t = 1 ms, the distribution of the strain field appears relatively uniform. The corresponding grayscale histogram demonstrates a "broad and low" profile, with a peak frequency of 0.0502 occurring at $k = 117$, indicating that most grayscale levels are concentrated in the mid-range of the grayscale interval.

At t = 3 ms, as shown in Fig 7(B), the specimen experiences the stress wave effects of $CO_2$ gas, leading to a strain concentration zone and the formation of primary cracks around the borehole. The grayscale histogram shifts towards higher grayscale levels, with a peak frequency of 0.0846 occurring at $k = 18$.

By t = 5 ms, demonstrated in Fig 7(C), the specimen undergoes the splitting action of $CO_2$ gas, causing the strain concentration zone to extends upward. The peak frequency in the grayscale histogram increases to 0.0913, with the strain field grayscale predominantly concentrated in the 0–100 range.

At t = 9 ms, shown in Fig 7(D), the generated gas further expands the cracks, and the strain localization zone rapidly enlarges. The grayscale histogram evolves to a "narrow and high" profile with a peak frequency of 0.1626, concentrating the strain field grayscale levels in the 0–30 range.

Table 2. Texture feature parameters.

| Formula | Meaning |
|---|---|
| $ASM = \sum_{i=0}^{255} \sum_{j=0}^{255} P(i,j,h,\theta)^2$ | Gray distribution uniformity |
| $CON = \sum_{i=0}^{255} \sum_{j=0}^{255} (i-j)^2 P(i,j,h,\theta)$ | Image sharpness and texture furrow depth |
| $COR = \dfrac{\sum_{i=0}^{255} \sum_{j=0}^{255} \left[ (ij)P(i,j,h,\theta) - \mu_x \mu_y \right]}{\sigma_x^2 \sigma_y^2}$ | The local gray similarity of the image |
| $ENT = -\sum_{i=0}^{255} \sum_{j=0}^{255} P(i,j,h,\theta) \log_2 P(i,j,h,\theta)$ | Randomness of texture |

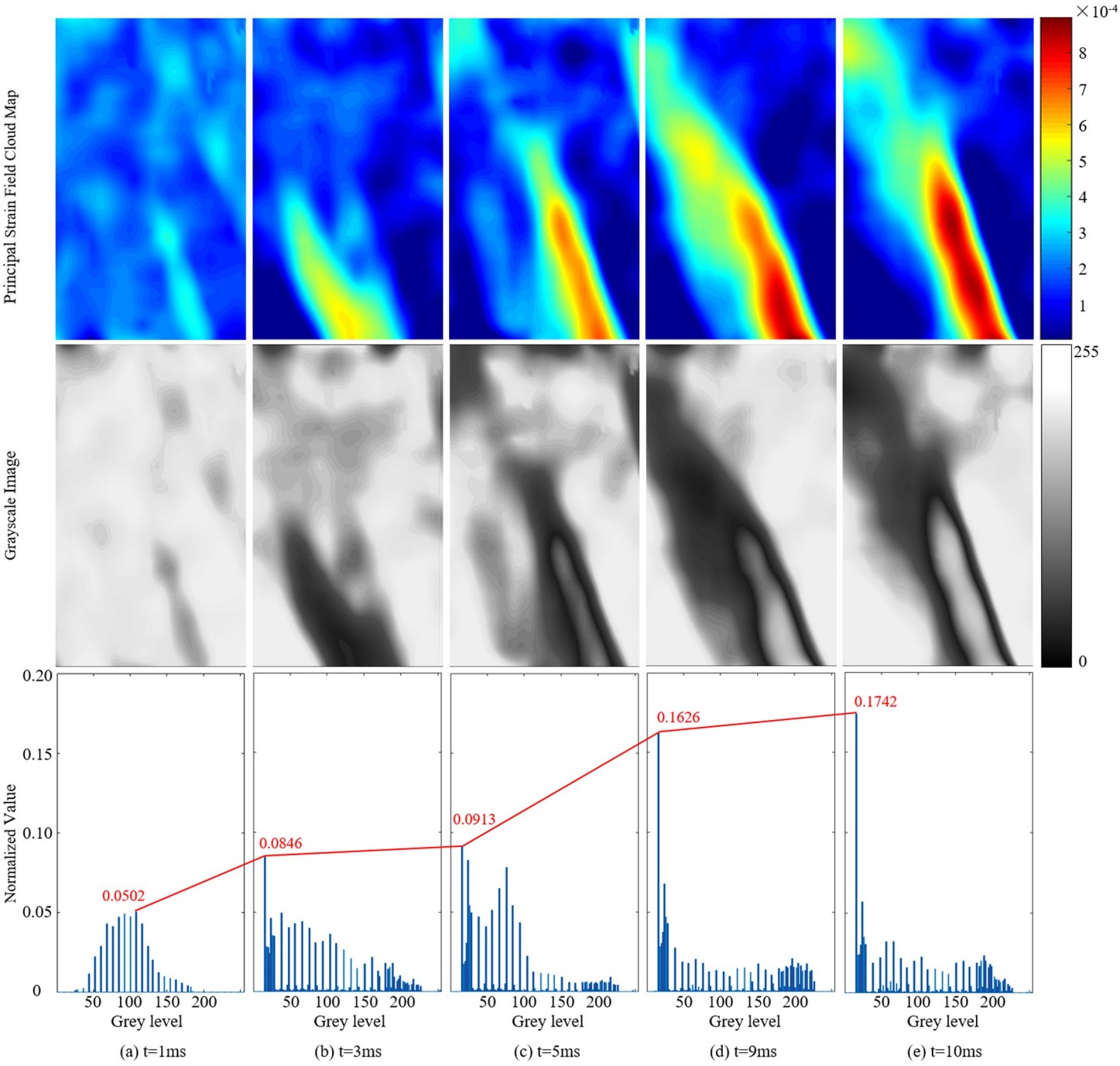

**Fig 7. Principal strain field, grayscale image and gray histogram of the sample during $CO_2$ blasting.**

Finally, at t = 10 ms, as depicted in Fig 7(E), the gas reflects off the damage surface, creating a radial jet that further stretches and damages the crack. The strain localization zone penetrates through the specimen. The corresponding grayscale histogram remains relatively unchanged, with the peak frequency increasing to 0.1742.

## Evolution law of texture feature parameters

For the extraction of texture feature images, offset parameters in different orientations (0˚, 45˚, 90˚, 135˚) were utilized to compute their respective GLCM. The features parameters were then averaged, with the results presented in Fig 8.

Fig 8(A) illustrates that at the onset of the $CO_2$ blasting, the materials surrounding the borehole were compacted by gas pressure, leading to a uniform primary strain texture distribution. At 3 ms, a strain concentration area emerged with high gray levels concentrated around the gas blasting tube. The ASM increased to 0.2582, signifying that the stress exceeded the tensile strength of the sample, initiating damage. The sample developed a main crack, forming an exhaust channel. After a brief $CO_2$ release, the gas refilled the fracture space, causing a temporary reduction in the strain localization zone before its subsequent expansion. The ASM rapidly declined to 0.1220 and then gradually increased. By 9 ms, the explosive gas further propagated the crack, enlarging the exhaust channel, with the ASM reaching 0.2032 and subsequently decreaseing to 0.1734.

As depicted in Fig 8(B), between 0–3 ms, the texture of the sample image transitioned from uniformly blurred to clear and distinct. During this phase, the CON uniformly increased. At 3 ms, a strain concentration area and a main crack emerged in the sample, leading to a decrease in CON from 0.2298 to 0.1987. Beyond 3 ms, as the strain localization zone gradually expanded with the emergence of a few low gray levels, the CON slowly decreased. After 9 ms, the CON abruptly rose from 0.0677 to 0.1002, corresponding to the sample's crack propagation under the continuous of $CO_2$ gas, resulting in a clear texture with deeper grooves.

Fig 8(C) shows that following the initiation of $CO_2$ blasting, the primary strain gray level gradually shifted from a uniform to a higher gray level distribution, with a generally decrease in COR. Before 3 ms, the emergence of the strain concentration area caused the COR to rapidly drop from 0.2362 to 0.0771. After 3 ms, as the strain localization zone expanded, the rate texture correlation decrease slowed. At 5 ms, the strain concentration area reduced, leading to a more dispersed strain distribution and a further decrease in COR. At 9 ms, the strain distribution concentrated in the main crack area, represented in the grayscale image as low gray levels in this region, and the COR slightly increased to 0.0502. Post 9 ms, with stress release and a more dispersed strain distribution, the COR decreased.

As demonstrated in Fig 8(D), following the blasting initiation, the ENT on the sample surface gradually accumulated. At 3 ms, stress concentration caused a shift in the image gray level frequency from almost equal to predominantly higher gray levels, leading to a decrease in image texture randomness and a sharp drop in ENT to 1.8674. From 3–9 ms, as the strain localization zone expanded, the image texture became more uniform, and the ENT generally declined. At 9 ms, a macroscopic crack formed in the sample due to explosive gas action, with the image gray level concentrating in the 0–30 range and uniformly distributed within the 31–255 gray level range, resulting in an increase in ENT, from 2.3183 to 2.4002.

## Mechanism and identification of precursor points for texture feature parameter changes

The texture feature parameters-time curves of the samples demonstrated distinct regularity and trends during the $CO_2$ blasting process. Analysis of the experimental data revealed that the texture feature parameters undergo changes at 3ms and 9ms. When correlated with the $CO_2$ blasting gas pressure distribution model, it was observed that at an initial pressure of 1.5MPa, the pressure in the borehole did not increase significantly at 3ms, indicating a propensity for rock fracturing. Post-fracture, gas escape and changes in borehole volume rendered the

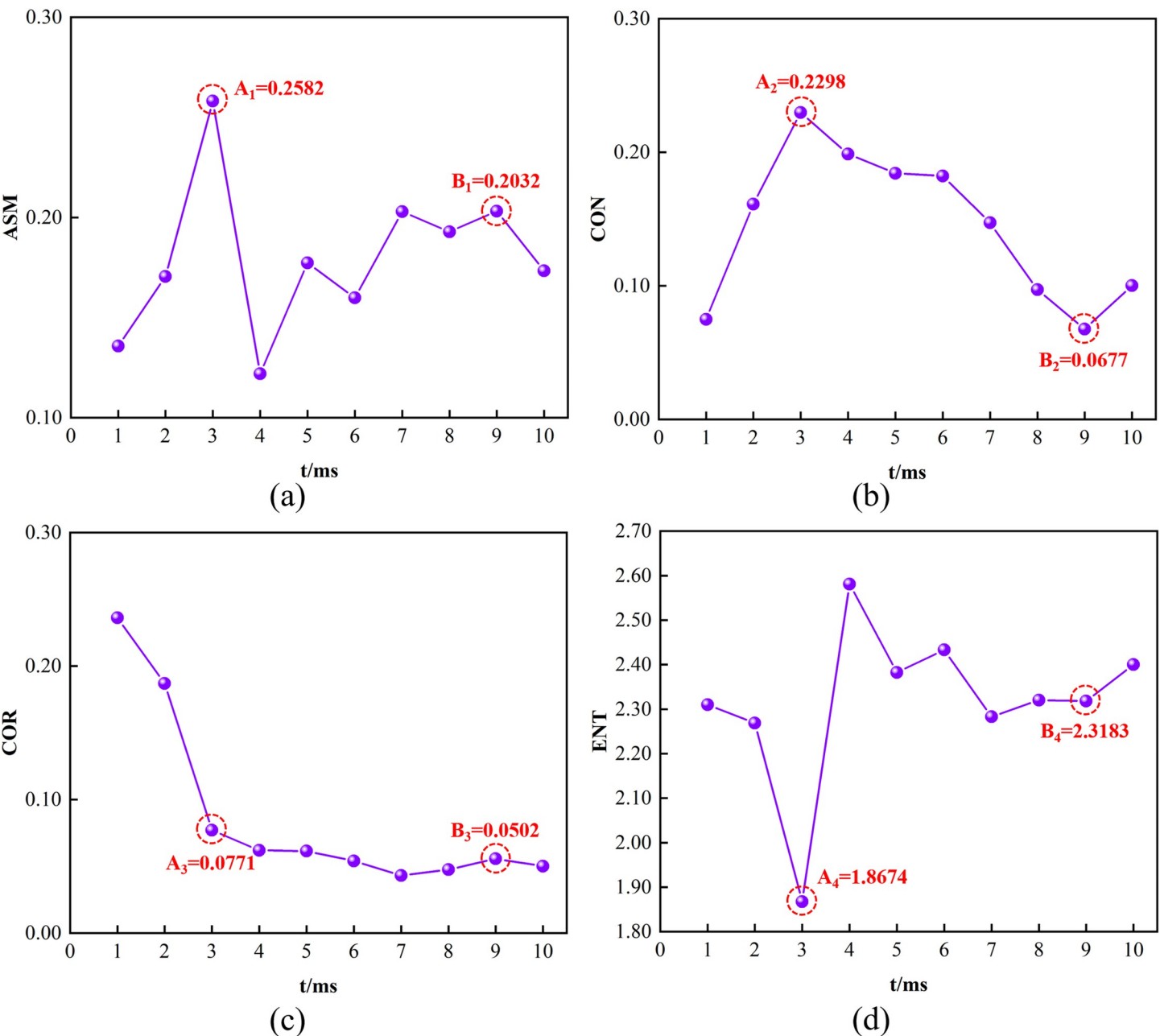

**Fig 8. Texture feature parameters-time curve of the $CO_2$ blasting process.**

model ineffective for pinpointing crack propagation timing. However, the second mutation of the texture feature parameters served as a reliable indicator for this purpose.

Based on the appearance of the main crack (point A in Fig 8), the texture feature parameters-time curves can be categorized into three distinct trends: abrupt increase, abrupt decrease, and rate reduction. Specifically, the ENT corresponds to the abrupt increase category, the ASM and CON to the abrupt decrease category, and the COR to the rate reduction category. The mutations in these parameters are attributed to the incidence of gas stress waves, where weak layers around the borehole first reach their compressive strength, generating main cracks

and forming exhaust channels. This process is reflected in the strain field as strain concentration areas around the borehole, leading to abrupt changes in the texture features of the strain cloud map. The rationale for categorizing the texture feature parameters into these trends is as follows:

1. ENT is positively correlated with the unevenness of local gray level distribution. A more uneven distribution results in a higher ENT value. At 3ms, when the strain concentration area emerges, low grayscale values are evenly distributed in the main crack direction, and high grayscale values in other areas, resulting in the lowest ENT value. Post 3 ms, as the explosive gas exerts a splitting effect on the crack tip, the grayscale distribution becomes uneven, causing a sudden increase in ENT.

2. ASM reflects the uniformity of image gray level distribution and texture coarseness, while CON indicates image clarity and the depth of texture grooves. Thus, upon sample cracking, these parameters exhibit an abrupt decrease.

3. CON is negatively correlated with local texture changes. Prior to coal rock cracking, the strain concentration area around the borehole emerges from nothing, with significant local changes leading to a rapid decline in CON. After cracking, the strain concentration area gradually expands, manifesting as a slowdown in the rate of CON reduction.

After the formation of the main crack, the explosive gas splits the crack tip, resulting in the gradual expansion of the strain localization zone. This phase is characterized in the strain field by the deepening of texture grooves in the strain cloud map, and an abrupt increase in CON (point B in Fig 8). The mutation patterns of other feature parameters are analogous to those observed at point A.

## Discussion

The main strain field grayscale and texture feature parameters can quantitatively characterize the coal rock cracking and propagation during the $CO_2$ blasting process and determine the times of coal rock cracking and propagation. The goal is to improve blasting permeability and gas extraction efficiency while avoiding environmental and safety issues caused by $CO_2$ misuse. The study provides a theoretical basis for enhancing gas extraction from coal seams. However, due to the limitations of simulation experiment conditions, a limitation of this study is that it solely employed uniaxial loading. The findings can be extrapolated to scenarios involving the fracturing of shallowly buried coal and rock, as well as coal rock containing numerous vertical faults, residual coal pillars, inclined coal seam slopes, and analogous rock-like materials without confining pressure. Nonetheless, for a more extensive range of applications, it becomes imperative to account for confining pressure within a closed-box environment, alongside the ramifications of $CO_2$ adsorption by coal. Additionally, the coal seam's adsorption characteristics, mechanical properties, and relevant conditions such as gas concentration should be considered to provide better theoretical guidance for $CO_2$ blasting permeability enhancement technology.

## Conclusion

The research delves into the evolution of strain field cloud map grayscale and texture features during the blasting process, focusing on identifying the precise moments of coal rock cracking and propagation. The key findings of the study are summarized as follows:

1. Utilizing the principle of stress superposition, a $CO_2$ blasting-induced coal rock fracture pressure model is developed. Additionally, employing the *R-K-S* equation, a $CO_2$ blasting

gas pressure distribution model is formulated. This model elucidates the variations in gas pressure within the borehole under different initial gas pressures over time.

2. A quantitative analysis of the $CO_2$ blasting strain field grayscale features is undertaken. The analysis of the grayscale histogram reveals a general shift from low to high grayscale levels, accompanied by a transition in shape from a "broad and low" to a "narrow and high" configuration, with a gradual increase in peak frequency. At 3 ms, the sample generates a primary crack, indicated by the grayscale peak frequency $P(18) = 0.0846$. Subsequently, at 9 ms, the crack further expands, and the grayscale peak frequency $P(18)$ rises to 0.1626.

3. The $CO_2$ blasting laboratory experiments demonstrate that the changes in the texture feature parameters-time curve of the sample strain field are indicative of the initiation and propagation of crack. These phenomena are represented in the strain field as the emergence and development of strain localization zones, leading to abrupt changes in the strain field grayscale and texture features. These alterations manifest as mutation segments on the curve. The principal strain field texture feature parameters-time curve exhibits mutations at 3 ms and 9 ms. In correlation with the laboratory experiments, these mutations correspond to the cracking time of the sample at 3 ms and the propagation time at 9 ms.

## Supporting information

**S1 File. The following is the MATLAB code for extracting texture feature parameters in RGB strain image.**
(DOCX)

**S2 File. This is the MTALAB code that converts the RGB strain image to grayscale image and plots the normalized gray histogram.**
(DOCX)

**S1 Data.**
(XLSX)

## Acknowledgments

Thanks to the test platform provided by the Key Laboratory of Western Mine Exploitation and Hazard Prevention of the Ministry of Education, the test was successfully completed, and data were obtained. Finally, the authors are also grateful to the anonymous reviewers for their constructive comments.

## Author Contributions

**Conceptualization:** Hongyu Pan, Haotian Wang.

**Data curation:** Haotian Wang, Kang Wang.

**Formal analysis:** Haotian Wang, Kang Wang.

**Funding acquisition:** Hongyu Pan, Tianjun Zhang.

**Investigation:** Bing Ji.

**Methodology:** Hongyu Pan, Haotian Wang.

**Project administration:** Hongyu Pan.

**Resources:** Tianjun Zhang.

**Software:** Haotian Wang.

**Supervision:** Hongyu Pan.

**Validation:** Hongyu Pan, Haotian Wang, Kang Wang.

**Visualization:** Tianjun Zhang.

**Writing – original draft:** Haotian Wang, Kang Wang.

**Writing – review & editing:** Hongyu Pan, Haotian Wang, Kang Wang.

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
