## [Decision Letter · Decision Letter 0]

17 Aug 2023

PONE-D-23-21600Experimental Study of Precursory Features of CO2 Blasting-induced Coal Rock Fracture based on Grayscale and Texture AnalysisPLOS ONE

Dear Dr. Wang,

Thank you for submitting your manuscript to PLOS ONE. After careful consideration, we feel that it has merit but does not fully meet PLOS ONE’s publication criteria as it currently stands. Therefore, we invite you to submit a revised version of the manuscript that addresses the points raised during the review process.

We look forward to receiving your revised manuscript.

Kind regards,

Ruchi Agrawal

Academic Editor

PLOS ONE

“The research study was successfully performed with contributions from all authors, and all authors approved the publication of the paper. The data to support the findings of this study are available from the corresponding author upon request. The authors declare that there are no conflicts of interest regarding the publication of this paper. This research was financially supported by The National Natural Science Foundation of China (52274226). The main research idea and manuscript preparation were contributed by Hongyu Pan; Haotian Wang contributed to the manuscript preparation and performed the correlative experiment. Kang Wang gave several suggestions from the industrial perspectives. Bing Ji, and Tianjun Zhang assisted in finalizing the research work and manuscript. Thanks to the test platform provided by the Key Laboratory of Western Mine Exploitation and Hazard Prevention of the Ministry of Education, the test was successfully completed, and data were obtained. Finally, The authors are also grateful to the anonymous reviewers for their constructive comments.”

“This research was financially supported by The National Natural Science Foundation of China (52274226).The funder had important role in writing – review & editing of the manuscript.”

Reviewers' comments:

Reviewer's Responses to Questions

**Comments to the Author**

1. Is the manuscript technically sound, and do the data support the conclusions?

Reviewer #1: No

Reviewer #2: Yes

Reviewer #3: Yes

Reviewer #4: Yes

Reviewer #5: Yes

Reviewer #6: Partly

2. Has the statistical analysis been performed appropriately and rigorously? 

Reviewer #1: Yes

Reviewer #2: Yes

Reviewer #3: Yes

Reviewer #4: Yes

Reviewer #5: Yes

Reviewer #6: N/A

3. Have the authors made all data underlying the findings in their manuscript fully available?

Reviewer #1: No

Reviewer #2: Yes

Reviewer #3: Yes

Reviewer #4: Yes

Reviewer #5: Yes

Reviewer #6: No

4. Is the manuscript presented in an intelligible fashion and written in standard English?

Reviewer #1: Yes

Reviewer #2: Yes

Reviewer #3: Yes

Reviewer #4: Yes

Reviewer #5: Yes

Reviewer #6: No

5. Review Comments to the Author

Reviewer #1: Many similar works have been done and no new understanding was gained. So this paper is lack of novelty to publish in PLOS ONE.

This study investigates the CO2 blasting permeability enhancement technology, an effective method for

improving coal seam permeability. To understand the deformation and fracture mechanisms of coal and rock

during CO2 blasting and identify their precursor characteristics, a CO2 blasting- induced coal rock fracture

pressure model, and a gas pressure distribution model were established. A self-developed CO2 gas explosion

test platform and a non-contact full-strain field measurement analysis system were employed. Briquette

samples were subjected to gas explosion tests under experimental conditions with an axial pressure of 1.0 MPa

and gas pressures of 0.5 MPa, 1.0 MPa, and 1.5 MPa. The principal strain field on the sample surface was

obtained. The quantitative analysis of the grayscale and texture characteristics’ evolution of the principal strain

field on the sample surface during CO2 blasting was conducted to identify the precursor characteristics of coaland rock cracking and re-expansion.

Reviewer #2: The research content of this manuscript is rich and easy to understand on the rock fracturing, but there is still room for improvement. In my opinion, it could be a very good article with some revision and organization. To help improve the quality of this manuscript, I have made some comments below.

1. The parameters appearing for the first time in the formula part of the article should indicate the unit.

2. the experimental equipment in the absence of authorization should be hidden equipment name.

3. The bar chart in Fig 4 should indicate the meaning of the ordinate.

4. The conclusion has repeated semantic parts, which can be more concise.

5. The word "re-expansion" appeared for the first time in the abstract, and there was no crack growth before, so it is suggested to modify it to avoid ambiguity.

6. In the experimental section where the following was mentioned, “Due to the large number of microcracks in the notched specimen, …of the specimens.” Although the core drilling technique does not ensure that the characteristics will be the same, it still can represent the natural formation of coal on a laboratory scale. In any case, coal is heterogeneous by nature and preparing the coal samples using certain ratios of elements, I feel removes the natural formation of coal and eliminates the possibility of replicating the actual coal rock. Please explain it.

Reviewer #3: This paper on CO2 blasting is interesting and easy to understand, it has certain significance and value for rock fracturing. The method adopted is novel, but it can be improved. In my opinion, if the article is revised and organized, it can become a very good article. To help improve the quality of this manuscript, I put forward some suggestions below：

1. In the introduction, the advantage of using the GLCM to deal with images may be added.

2. In line 191, the results were normalized, and it should explain how to complete the process.

3. This paper studied the gray and texture features under different gas explosion pressure experimental conditions. However, the author should explain which equipment can control the initial gas explosion pressure in the experimental system in Fig. 1.

4. The author should keep the consistency of words (such as strain field and CO2 blasting).

Reviewer #4: The author studied the evolution law of gray and texture features of the strain field cloud image during the blasting process, determined the time of coal rock cracking and re-expansion, and achieved good results.The conclusion has reflected their analysis and the manuscript is in a proper format to understand the content very well. Recommended for publication in journals after minor repairs.

Questions to Authors:

1. The value, importance and originality of the paper should be clearly stated in the introduction section.

2. Eliminate multiple references. After that please check the manuscript thoroughly and eliminate all the lumps in the manuscript. This should be done by characterising each reference individually This can be done by mentioning 1or 2 phrases per reference to show how it is different from the others and why it deserves mentioning.

3. Carefully approve the symbols used in the formula and make a reasonable explanation, such as the meaning of r in Eq.5.

4 . The results section should clearly state how the data were analyzed and what statistical tests were used.

5. The abstract of the article needs to be revised, and the importance of the article should be highlighted.

6. It would be beneficial to discuss the limitations and potential areas for improvement of the proposed model.

7. The results are not discussed in depth. I feel that the authors could explore their findings in more details. Furthermore, there is no validation study of the model with experimental data. How do the authors justify?

Reviewer #5: Please see my comment in the attachment. The research establishes a CO2 blasting-induced coal rock fracture pressure model, a gas pressure distribution model, and examines the evolution of strain field characteristics during the blasting process. From my evaluation, it is an interesting topic and can inspire the investigation of this area. Overall, this paper is well prepared and well written. Overall, considering the novelty and contribution, this paper can be considered after minor revision.

Reviewer #6: This paper reports experimental data on the surface principal strain field of briquette samples exposure to CO2 blusting using a grayscale and texture analysis method. The research topic is interesting, but the manuscript is not reader-friendly, as it is quite difficult to follow details. It is confusing that the role of theoretical part, i.e. what is the link between the theory and experiments? The following details might be helpful to improve the manuscript.

1. The section of theory looks messy. The equations were not well described and explained, as well as the symbols used in the formulars were not well defined.

2. Line 14 states three sets of experiments performed, but the analysis of grayscale histograms, principal strain cloud maps, and texture feature parameters were found only for one set of experimental results. What is the significance of the remaining two sets of experiments?

3. The parameter P of formula 3 is not defined;The explanation of parameters in lines 87-90 would be better if a schematic diagram could be made to illustrate the effect; The r in formula 5 is not defined; The derivation process of Formula 7 should be more readable to readers; Tr in formula 10 is not defined; The second Y=Nx-1 in formula 17 should be Y=Ny-1? ...

4. The figures need improve. Figure 1, 2D plan view might be better to read the data; Fig.4 and Fig.6 need more specific descriptions, e.g. Fig.6c, there is insufficient analysis of the relationship between the trend of curve changes and the changes in the experimental formula. The trend of changes in the second half of the journey does not match the A and B charts. Why did the trend of changes increase before 9ms? This is opposite to the trend of changes before 3ms, which is illogical.

5. Format errors in References 4 and 25...

6. Lines 269-276 do not mention the basis for dividing the three types of trends. And regardless of the slope or extreme point before and after point A, the four charts cannot fully correspond to the three trend types.

7. Lines 277-278, it is difficult to see the positive correlation, and the entropy value should decrease with the formation of the strain concentration area?

8. Lines 282-284 seem not closely related to the research content in the previous text.

9. No clue for how to determine the cracking moment and expansion moment.

10. Experimental details, such as data reduction, should be added.

6. PLOS authors have the option to publish the peer review history of their article (what does this mean?). If published, this will include your full peer review and any attached files.

Reviewer #1: No

Reviewer #2: No

Reviewer #3: No

Reviewer #4: No

Reviewer #5: **Yes: **Changtai Zhou

Reviewer #6: No

---

## [Decision Letter · Decision Letter 1]

17 Oct 2023

PONE-D-23-21600R1Experimental study of precursory features of CO2 blasting-induced coal rock fracture based on grayscale and texture analysisPLOS ONE

Dear Dr. Wang,

Thank you for submitting your manuscript to PLOS ONE. After careful consideration, we feel that it has merit but does not fully meet PLOS ONE’s publication criteria as it currently stands. Therefore, we invite you to submit a revised version of the manuscript that addresses the points raised during the review process.

We look forward to receiving your revised manuscript.

Kind regards,

Ruchi Agrawal

Academic Editor

PLOS ONE

Reviewers' comments:

Reviewer's Responses to Questions

**Comments to the Author**

1. If the authors have adequately addressed your comments raised in a previous round of review and you feel that this manuscript is now acceptable for publication, you may indicate that here to bypass the “Comments to the Author” section, enter your conflict of interest statement in the “Confidential to Editor” section, and submit your "Accept" recommendation.

Reviewer #2: All comments have been addressed

Reviewer #4: All comments have been addressed

Reviewer #7: All comments have been addressed

Reviewer #8: (No Response)

Reviewer #9: (No Response)

Reviewer #10: (No Response)

Reviewer #11: All comments have been addressed

Reviewer #12: (No Response)

Reviewer #13: (No Response)

Reviewer #14: (No Response)

Reviewer #15: All comments have been addressed

Reviewer #16: All comments have been addressed

Reviewer #17: (No Response)

2. Is the manuscript technically sound, and do the data support the conclusions?

Reviewer #2: Yes

Reviewer #4: Yes

Reviewer #7: Partly

Reviewer #8: Yes

Reviewer #9: (No Response)

Reviewer #10: (No Response)

Reviewer #11: Yes

Reviewer #12: Yes

Reviewer #13: No

Reviewer #14: Partly

Reviewer #15: Yes

Reviewer #16: Yes

Reviewer #17: Partly

3. Has the statistical analysis been performed appropriately and rigorously? 

Reviewer #2: Yes

Reviewer #4: Yes

Reviewer #7: N/A

Reviewer #8: Yes

Reviewer #9: (No Response)

Reviewer #10: (No Response)

Reviewer #11: Yes

Reviewer #12: Yes

Reviewer #13: No

Reviewer #14: No

Reviewer #15: (No Response)

Reviewer #16: Yes

Reviewer #17: No

4. Have the authors made all data underlying the findings in their manuscript fully available?

Reviewer #2: Yes

Reviewer #4: Yes

Reviewer #7: Yes

Reviewer #8: Yes

Reviewer #9: (No Response)

Reviewer #10: (No Response)

Reviewer #11: (No Response)

Reviewer #12: Yes

Reviewer #13: No

Reviewer #14: No

Reviewer #15: (No Response)

Reviewer #16: Yes

Reviewer #17: Yes

5. Is the manuscript presented in an intelligible fashion and written in standard English?

Reviewer #2: Yes

Reviewer #4: Yes

Reviewer #7: Yes

Reviewer #8: Yes

Reviewer #9: (No Response)

Reviewer #10: (No Response)

Reviewer #11: (No Response)

Reviewer #12: Yes

Reviewer #13: No

Reviewer #14: Yes

Reviewer #15: (No Response)

Reviewer #16: Yes

Reviewer #17: No

6. Review Comments to the Author

Reviewer #2: (No Response)

Reviewer #4: (No Response)

Reviewer #7: Manuscript entitled ‘Experimental study of precursory features of CO2 blasting-induced coal rock fracture based on grayscale and texture analysis’ discussed the possibility of applying GLCM method to identify the precursor characteristics of coal rock cracking and crack propagation. The study suffers some fundamental flaws that hinder the acceptance of this paper, detailed comments and suggestions are attached below for the authors’ reference.

1.In the introduction section, too many Chinese references, and most literature are also from Chinese authors, it is suggested that the authors shall expand their reading scope and exclude some Chinese references.

2.L102 in P10, deep and tight coal seams? what does it mean? Coal seams with gas that is difficult to be extracted?

3.Fig.1 is an existing theory, reference shall be added here.

4.The method of obtaining Fig. 2 shall be explained, current layout of the curves is wield.

5.The experimental design in this study had serious errors, the setup was too simply, only one specimen was prepared for each group, it would be better if the authors add some explanations here why only one specimen was selected from each group.

6.L219 in P17, why only the upper half of the specimen was selected for analysis?

7.L221 in P17, how did the authors control the pressure and set the value as 1.5 MPa?

8.The authors stated that ‘the grayscale is quantized from 256 levels into 64 levels’, however, the red and the blue color in original figures had been altered to all white color, see Fig. 5. How did the authors consider the errors this degradation could incur?

Reviewer #8: This article uses gray and texture analysis methods, combined with CO2 blasting, and through theoretical analysis and laboratory experiments, it shows the process of CO2 blasting-induced coal rock fracture, and obtains the precursory characteristics of cracks from initiation to expansion. The scientific nature of the work is good. I think this research is meaningful, but this article needs to be minor revisions so that it can be published.

1. Compared with the original manuscript, the grammatical problems of the revised manuscript have been greatly improved, but there are still some small problems that need to be modified.

2. Previous studies on CO2 induced fracturing have done a lot of work. It is suggested that the author should evaluate objectively in the citation, and highlight the characteristics of the research and the difference from the existing research.

3. It is recommended to upload a clearer version of Figure 7 for publication.

4. How many experiments were performed? Please add description in experiment section.

5. Were any saturation performed before any experiment?

6. Conclusion The description is too complicated, and it is suggested to reorganize the language.

7. It is suggested that the last paragraph of the conclusion be re-summarized into a chapter and placed before the conclusion, which will be clearer. Just like this:

Discussion

“The main strain field grayscale and texture features parameters can quantitatively characterize the coal rock cracking and propagation during the CO2 blasting process and determine the times of coal rock cracking and propagation. The goal is to improve blasting permeability and gas extraction efficiency while avoiding environmental and safety issues caused by CO2 misuse. The study provides a theoretical basis for enhancing gas extraction from coal seams. However, due to the limitations of simulation experiment conditions, a limitation of this study is that it solely employed uniaxial loading. The findings can be extrapolated to scenarios involving the fracturing of shallowly buried coal and rock, as well as coal rock containing numerous vertical faults, residual coal pillars, inclined coal seam slopes, and analogous rock-like materials without confining pressure. Nonetheless, for a more extensive range of applications, it becomes imperative to account for confining pressure within a closed-box environment, alongside the ramifications of CO2 adsorption by coal. Additionally, the coal seam's adsorption characteristics, mechanical properties, and relevant conditions such as gas concentration should be considered to provide better theoretical guidance for CO2 blasting permeability enhancement technology.”

Reviewer #9: 本研究重点研究了CO2注入过程中应变场云图像灰度和织构特征的演变。构建了CO2爆破引起的煤岩断裂压力模型和CO2爆破气体压力分布模型。定量研究了CO2爆破应变场的灰度特征，分析了织构特征参数随时间的变化规律。该研究旨在提高鼓风渗透性和气体提取效率，同时避免与二氧化碳滥用相关的环境和安全问题。虽然我发现了一些改善工作的问题，如下所示

2.手稿中使用的英语太差了，以至于无法理解一句话的含义。

1.手稿中提到的方程式都是已知的和预先存在的。在这方面也没有新的贡献。

2.分析不足以验证结果。

3. CO4压裂与水力压裂在压裂压力和裂缝网络复杂性方面的比较如何？

2、灰度和织构分析在确定煤岩开裂再展开前驱特征方面得出的结论是什么？

5. 与CO6注入相关的研究的目的是什么？

2. 研究中如何获得应变场的灰度和织构参数？

7. 我没有看到任何新颖的东西可以提交给影响力较低的期刊考虑。

Reviewer #10: In this study, the model of coal rock fracture pressure caused by CO2 blasting and the model of gas pressure distribution caused by CO2 blasting are constructed. The gray-scale characteristics of CO2 blasting strain field are quantitatively studied, and the variation of texture parameters with time is analyzed. The aim is to improve blast permeability and gas extraction efficiency, while reducing environmental and safety concerns associated with CO2 abuse. Therefore, this study is not very innovative and I do not think it is suitable for publication in PLOS ONE journal. The following suggestions are given to the author for reference:

1. The language in the manuscript is difficult to understand, and it is impossible to read the meaning expressed in the article, and it is recommended to improve the use of language.

2. Most of the references in this paper are Chinese scholars, so we should make appropriate reference to foreign scholars to enrich the basic research of this article.

3. The model constructed and the formula used in this article are already ubiquitous and not innovative.

4. The content of the article is single, there are few diagrams, and it is not convincing.

5. The analysis of the conclusions obtained in the article is too simple to prove the topic of the article, and it is recommended to further condense the conclusions.

6. How to obtain the grayscale and texture parameters of the strain field in the study are not reflected in the paper, and it is recommended that the authors add further.

7. The results are not of publication value, as conclusions are usually not general and some conclusions are debatable or even wrong.

8. The overall content of the article is not innovative and of poor quality, and it is recommended to turn to journals with low impact.

Reviewer #11: The content of the article is complete and the structure is clear, and it has been greatly improved after revision. I would like to ask the author to revise some details, and suggest that it be published after minor revision.

(1) Line 255, the abbreviation ASM, it is recommended that the full name be Angular Second Moment instead of energy, which can be confusing for readers outside the field.

(2) Line 256, Correlation is abbreviated as CON, while CORRLN is written in Table 2, which needs to be unified.

(3) In the manuscript, we can see that the author sets three groups of samples, and only analyzes the case of 1.5MPa in the results and discussion. Why choose the case of 1.5MPa for analysis? Why not choose 0.5Mpa or 1.0MPa? I think the other two groups should be properly analyzed even as control groups. Please reply.

Reviewer #12: In this study, a series of CO2 blasting tests were conducted on coal-like material to investigate the precursory features of coal rock cracking and crack propagation. The tests were carried out under different initial gas pressure conditions (0.5 MPa, 1.0 MPa, and 1.5 MPa). The non-contact full-strain field measurement analysis system was used to analyze the evolution law of strain field and texture features. The manuscript can be accepted if the authors consider the following suggestions:

1) The authors have provided a comprehensive review of previous research on CO2 blasting in the second paragraph of the Introduction (Line 52-76). However, the literature review is too lengthy and should be condensed to highlight the similarities and differences.

2) If the theory and calculation formulas in this study are based on existing literature, it is recommended to include references.

3) It is recommended to include a sample diagram and specify the size of the gas blast tube or the sample notch.

4) What is the rationale for establishing different initial gas pressure conditions? Does it hold specific practical engineering significance?

5) In Figure 5, the principal strain field, grayscale image, and gray histogram exhibit a clear trend. How do the results of the other two groups (when the initial gas pressure is 0.5 MPa and 1.0 MPa) compare?

Reviewer #13: I can't see any novelties and it can be submitted to lower impact journals for consideration. The manuscript is not suitable for consideration in its present format. In this paper, I have found some issues towards the betterment of the work which are listed below:

1.English used in the manuscript is very poor to understand the meaning of one sentence.

2.The equations mentioned in the manuscript are all known and pre-existing. Novel contribution in respect to that is also missing.

3.Analysis is not sufficient to validate the results.

4.A comprehensive comparative analysis is missing to proof the outperformance of the proposed method.

5.Benchmark testing as well as statistical analysis of the results is also missing in the work.

6.Author has to perform more detail comparative analysis with the recent state-of-the-art method to validate the results.

Reviewer #14: The idea is not original enough and the amount of experimentation reflects too little.Inadequate data analysis and presentation,also the structure of the article still needs to be adjusted.After our careful consideration, it does not fit PLOS ONE's position.

Reviewer #15: The paper entitled “Experimental study of precursory features of CO2 blasting-induced coal rock fracture based on grayscale and texture analysis” uses GLCM to extract the grayscale and texture features of the evolution of grayscale and textural features, quantitatively analyze the grayscale and texture features evolution of the main strain field on the sample surface during the CO2 blasting process, and identify the precursor characteristics of coal rock cracking and crack propagation. This research is interesting, and the developed test platform has specific application significance, which can provide a reference for the CO2 blasting mechanism. I think this manuscript can be accepted after the following revision.

1. The axes and labels of some pictures are not clear.

2. Conclusion, nouns that appear previously should be consistent. For example, “Matlab” should be capitalized, or the previous text should be changed to lowercase.

3. Line 135-138, parameters should be italicized to be consistent with the formula.

4. Line 149, the word “borepore” should be spelled incorrectly, “borehole” is used in the article, please check carefully. Line 227, Figure 5b should be revised to Fig 5b. Please check other similar errors

5. When several quantities with units appear together, add the unit only after the last one. For example, “0.5 MPa, 1.0 MPa, and 1.5 MPa”, please check the full text and modify it.

Reviewer #16: I have reviewed an article “Experimental study of precursory features of CO2 blasting-induced coal rock fracture based on grayscale and texture analysis”. The manuscript falls within the scope of the journal. This manuscript can be considered for publication after the revision suggested below:

1. Line 212: “Grayscale is an integer … brightness levels”. It does not reflect the advantages of grayscale analysis and it is recommended to modify it.

2. The font size in the image needs to be adjusted.

3. Please check whether the parameters in the formula part are all annotated and correspond one to one.

4. Line 273: “As shown in Fig 7(b), from 0 ms to 3 ms …”. When describing this time period, only the last unit needs to be retained. The same error appears in this section, please check and correct it.

5. Conclusions needs to be streamlined.

Reviewer #17: Dear Editors and Authors:

I am honored to be able to review this paper. This paper studies the precursor information of precursory features of CO2 blasting-induced coal rock fracture based on grayscale and texture analysis. To my knowledge, the engineering background of the research content in the paper is very complex, and the development of cracks requires at least three stages: guided cracks, initial cracks, and crack expansion. However, the conclusions of this paper are lack of convincing and there are confusions in the paper that need to be clarified. The following are suggestions for revision:

1. English grammar is quite chaotic, and English translation requires professional proofreading.

2. During the coal rock blasting process, similarity ratio, materials, and model size have a significant impact on the research results. The development of cracks is a very complex process. It is questionable to what extent such a small model experiment can reflect the actual situation.

3. What is the connection between model testing and actual mines? How to consider the material and related rock mass structure of coal mines? Only using a single method to study precursor characteristics lacks innovation.

Suggest rejection

We appreciate for Editors/Authors’ warm work earnestly.Thank you and Best regards.

Yours sincerely,

Reviewer

7. PLOS authors have the option to publish the peer review history of their article (what does this mean?). If published, this will include your full peer review and any attached files.

Reviewer #2: No

Reviewer #4: No

Reviewer #7: No

Reviewer #8: No

Reviewer #9: No

Reviewer #10: No

Reviewer #11: No

Reviewer #12: No

Reviewer #13: No

Reviewer #14: No

Reviewer #15: No

Reviewer #16: No

Reviewer #17: No

---

## [Author Response · Author response to Decision Letter 1]

4 Dec 2023

We have uploaded the "Response to Reviewers" file to the system. Thank you very much for reviewing the manuscript and giving us the opportunity to revise and improve its quality. We have revised the article in terms of technology and writing. Please review it.

---

## [Decision Letter · Decision Letter 2]

21 Dec 2023

PONE-D-23-21600R2Experimental study of precursory features of CO2 blasting-induced coal rock fracture based on grayscale and texture analysisPLOS ONE

Dear Dr. Wang,

Thank you for submitting your manuscript to PLOS ONE. After careful consideration, we feel that it has merit but does not fully meet PLOS ONE’s publication criteria as it currently stands. Therefore, we invite you to submit a revised version of the manuscript that addresses the points raised during the review process.

Thanks to authors for revising the article. It is further suggested to discuss and corelate this article with the SDG goals defined by UN and the concept of Circular Economy. Following studies are suggested for reference purpose and may be cited (https://doi.org/10.1016/j.envres.2023.115523, https://doi.org/10.1007/s10570-023-05068-0, https://doi.org/10.1080/21655979.2023.2242124, https://doi.org/10.1016/B978-0-323-91187-0.00002-3).

We look forward to receiving your revised manuscript.

Kind regards,

Ruchi Agrawal

Academic Editor

PLOS ONE

Journal Requirements:

Reviewers' comments:

Reviewer's Responses to Questions

**Comments to the Author**

1. If the authors have adequately addressed your comments raised in a previous round of review and you feel that this manuscript is now acceptable for publication, you may indicate that here to bypass the “Comments to the Author” section, enter your conflict of interest statement in the “Confidential to Editor” section, and submit your "Accept" recommendation.

Reviewer #12: All comments have been addressed

Reviewer #16: All comments have been addressed

2. Is the manuscript technically sound, and do the data support the conclusions?

Reviewer #12: Yes

Reviewer #16: Yes

3. Has the statistical analysis been performed appropriately and rigorously? 

Reviewer #12: Yes

Reviewer #16: Yes

4. Have the authors made all data underlying the findings in their manuscript fully available?

Reviewer #12: Yes

Reviewer #16: Yes

5. Is the manuscript presented in an intelligible fashion and written in standard English?

Reviewer #12: Yes

Reviewer #16: Yes

6. Review Comments to the Author

Reviewer #12: The Comments that we proposed are all properly addressed. Now I'm sure that this revised manuscript can be accepted for publication in PLOS ONE.

Reviewer #16: I have carefully checked the revised manuscript. I am satisfied. I would recommend to publish this manuscript.

7. PLOS authors have the option to publish the peer review history of their article (what does this mean?). If published, this will include your full peer review and any attached files.

Reviewer #12: No

Reviewer #16: No

---

## [Author Response · Author response to Decision Letter 2]

24 Dec 2023

The “Response to Reviewers” file has been uploaded in the system

---

## [Editor Report · Decision Letter 3]

12 Jan 2024

Experimental study of precursory features of CO2 blasting-induced coal rock fracture based on grayscale and texture analysis

PONE-D-23-21600R3

Dear Dr. Wang,

We’re pleased to inform you that your manuscript has been judged scientifically suitable for publication and will be formally accepted for publication once it meets all outstanding technical requirements.

Kind regards,

Ruchi Agrawal

Academic Editor

PLOS ONE
---

## [Editor Report · Acceptance letter]

1 Feb 2024

PONE-D-23-21600R3 

PLOS ONE

Dear Dr. Wang, 

I'm pleased to inform you that your manuscript has been deemed suitable for publication in PLOS ONE. Congratulations! Your manuscript is now being handed over to our production team.

Kind regards, 

on behalf of

Dr. Ruchi Agrawal 

Academic Editor

PLOS ONE